# Optimization of Pulsed Laser Ablation and Radio-Frequency Sputtering Tandem System for Synthesis of 2D/3D Al_2_O_3_-ZnO Nanostructures: A Hybrid Approach to Synthesis of Nanostructures for Gas Sensing Applications

**DOI:** 10.3390/nano13081345

**Published:** 2023-04-12

**Authors:** Joselito Puzon Labis, Hamad A. Albrithen, Mahmoud Hezam, Muhammad Ali Shar, Ahmad Algarni, Abdulaziz N. Alhazaa, Ahmed Mohamed El-Toni, Mohammad Abdulaziz Alduraibi

**Affiliations:** 1King Abdullah Institute for Nanotechnology, King Saud University, Riyadh 11451, Saudi Arabia; 2Department of Physics and Astronomy, College of Science, King Saud University, Riyadh 11451, Saudi Arabia

**Keywords:** zinc oxide, aluminum oxide, pulsed laser deposition, laser ablation, radio-frequency magnetron sputtering

## Abstract

In this paper, a unique hybrid approach to design and synthesize 2D/3D Al_2_O_3_-ZnO nanostructures by simultaneous deposition is presented. Pulsed laser deposition (PLD) and RF magnetron sputtering (RFMS) methods are redeveloped into a single tandem system to create a mixed-species plasma to grow ZnO nanostructures for gas sensing applications. In this set-up, the parameters of PLD have been optimized and explored with RFMS parameters to design 2D/3D Al_2_O_3_-ZnO nanostructures, including nanoneedles/nanospikes, nanowalls, and nanorods, among others. The RF power of magnetron system with Al_2_O_3_ target is explored from 10 to 50 W, while the ZnO-loaded PLD’s laser fluence and background gases are optimized to simultaneously grow ZnO and Al_2_O_3_-ZnO nanostructures. The nanostructures are either grown via 2-step template approach, or by direct growth on Si (111) and MgO<0001> substrates. In this approach, a thin ZnO template/film was initially grown on the substrate by PLD at ~300 °C under ~10 milliTorr (1.3 Pa) O_2_ background pressure, followed by growth of either ZnO or Al_2_O_3_-ZnO, using PLD and RFMS simultaneously under 0.1–0.5 Torr (13–67 Pa), and Ar or Ar/O_2_ background in the substrate temperate range of 550–700 °C. Growth mechanisms are then proposed to explain the formation of Al_2_O_3_-ZnO nanostructures. The optimized parameters from PLD-RFMS are then used to grow nanostructures on Au-patterned Al_2_O_3_-based gas sensor to test its response to CO gas from 200 to 400 °C, and a good response is observed at ~350 °C. The grown ZnO and Al_2_O_3_-ZnO nanostructures are quite exceptional and remarkable and have potential applications in optoelectronics, such in bio/gas sensors.

## 1. Introduction

Inherently an n-type material (with 3.37 eV-bandgap and binding energy of 60 meV-exciton), zinc oxide (ZnO) is a remarkably stable and low-cost semiconductor that exhibits excellent optoelectronics and superior electrical properties [1]. ZnO has been, therefore, intensively and extensively explored for bio/gas sensing applications [1,2,3]. Because of its unique properties, it has been synthesized into different 2D/3D nanostructures, such as nanorods [4], nanoneedle/nanospikes [5], nanoflowers, [6], and other exceptional nanostructures [7]. With high surface-to-volume ratio, ZnO nanostructures have found their way into various applications in material science, such as in the area of gas/bio-sensing [1,6].

However, the task of designing, doping, and functionalizing 2D/3D ZnO nanostructures suitable for gas/bio sensing applications remains a technical challenge, as the nanostructures have to be directly synthesized and incorporated into the base substrate to create high-quality sensors with exceptional and rapid responses. Most synthesis methods involve indirect synthesis, such as spray pyrolysis via sol-gel (and other tedious techniques), followed by screen-plating on the substrate. This particular method was found to be ineffective due to adhesion issue [3,8]. 

As nanomaterials that are added directly onto the electrode, they are expected to have strong adhesion and improved electrical contact, which should result in high stability, high temperature tolerance and high response [9]. Therefore, a newer method must be developed that would directly incorporate the grown ZnO nanostructures into the sensor substrate. At the same time, we must design nanostructures that possess the unique capability of simultaneously functionalizing/doping the grown structures with other metals/compounds and thus transforming them into ZnO heteronanostructures.

The overall performance and characteristics of a good gas sensor depend on the following: the morphological structure of the material, the surface-to-volume ratio, the crystallinity of the material and its chemical and electronic properties, and the deposition technique used [6]. However, most sensors still suffer from low response, poor selectivity, and high operating temperatures [6,7]. As heteronanostructures, they are expected to improve the performance of gas sensor by facilitating catalytic activities, increasing oxygen adsorptions, and creating charge carrier depletion layers that produce larger modulation in resistance [10]; there is, therefore, a vital need to develop a tandem dual/co-deposition system that is capable of designing, synthesizing, and functionalizing heteronanostructures. 

In this regard, PLD offers a unique advantage over other growth methods, as PLD can directly and stoichiometrically transfer the target material into a substrate as a film or as nanostructures with the same material composition [11]. By optimizing the PLD parameters, interesting ZnO nanostructures could be grown that could have potentials in gas sensing applications [10,11]. Although PLD is capable of growing different 2D/3D ZnO nanostructures directly into the substrate, it has limitations in terms of doping as it can only utilize targets doped at different percentages for ablation. The pressed doped target may not have the desired percentages, as the dopants may not be homogeneously doped into the ZnO structures or matrix during the target preparation. Therefore, to enhance the gas sensing properties of the grown ZnO nanostructures, a new functionalizing/doping method has to be employed on physically/chemically doped materials in the ZnO structures.

RFMS, on the other hand, is a very good method of growing films and nanostructures. One of its potential uses comes from its ability to scale the growth process from laboratory to industrial scale; this is, on the other hand, one of the limitations of PLD [11]. Therefore, hybridizing PLD with RFMS makes these two physical vapor deposition systems into a unique system, one that is capable of simultaneously growing high-purity heteronanostructures while physically functionalizing/doping them with interesting other materials. The physical, electrical, and electronic properties of the resulting functionalized/doped heteronanostructures are expected to be promising in the area of gas sensing. The ability of this tandem system to individually control the chemical and phase composition of the resulting structures makes this system a very versatile tool for assisting the growth of nanostructures, since, in this process, the dopants are directly introduced into the plume plasmas, with a good chance of homogeneously functionalizing the ZnO. 

The tandem system of PLD and RFMS is expected, therefore, to be a well-suited tool for designing remarkable heteronanostructures for gas sensing applications [12]. The tandem of PLD and RFMS has indeed offered a viable and effective doping/functionalization method for designing and growing semiconducting oxide heteronanostructures [10]. Recently, this hybrid tandem system has been explored for use to design and grow doped nanostructures, coatings, and thin films. Shin and Son demonstrated co-deposition using PLD and RF sputtering to deposit Cu-doped ZnO and showed the potential applications of the dual system [13]. Voevodin et al. previously explored and demonstrated this dual deposition technique in 1996, which was early for the field, using a combination of PLD and magnetron sputtering to grow carbide- and diamond-like carbon films [14]. However, only very few reports have been published since then. This tandem set-up has been recently revived and explored to design and synthesize different nanostructures and multilayers of films. Very recently, Benetti et al. incorporated an RF system to enhance the pulsed laser ablation-deposition of TiO_2_ and Bi_2_FeCrO_6_ thin films [15]. Further, this hybrid was recently explored in growing Zn-Au films, where Au was found to be uniformly incorporated throughout the film without structural alteration to the crystallographic Wurzite feature of the ZnO [10]. More recently, Chrzanowska-Giżyńskaa et al. used a similar hybrid system of PLD and RFMS to deposit superhard coatings of tungsten boride [16]. This hybrid system has uniquely combined the highly energetic UV laser plasma (up to 1 keV) with that of the low-energy RF magnetron plasma (up to several eV); therefore, the combination, together with the geometrical features of RFMS inside the PLD system, offers flexibility in designing nanoheterostructures with properties suited for gas sensing applications [17]. 

Although the technique offers an advantage over other conventional methods, understanding the mechanism process and the physics of mixed plasma remains unknown and still under debate because the combined parameters have to be well optimized to produce 2D/3D ZnO heteronanostructures. Further, simultaneously generating two plasmas with PLD and RFMS remains a challenge and requires a comprehensive understanding of their dynamics. 

In this paper, the parameters of the hybrid tandem systems of PLD and RFMS are optimized to grow unique 2D/3D ZnO and Al_2_O_3_-ZnO heteronanostructures with high surface-to-volume ratios by understanding the mechanism of growth within the system, as well as their plasma dynamics for potential gas sensing applications.

## 2. Materials and Methods

In this study, the Neocera Pioneer 180 PLD system, equipped with a 248 nm KrF Laser Compex (Lambda-Physik, Compex 205, Santa Clara, CA, USA), was combined with a compact 1-inch RF magnetron sputtering system (AJA 100/300 MM3) to simultaneously grow 2D/3D ZnO and Al_2_O_3_-ZnO heteronanostructures. Figure 1 shows the schematic diagram of the tandem PLD and RFMS systems. Prior to deposition, the PLD system was pumped using a turbomolecular pump and a rotary pump up to a base pressure of ~10^−8^ Torr (~10^−6^ Pa). The excimer KrF laser was focused on the PLD target (99.999% purity, commercial 2-inch ZnO target, Kurt J. Lesker Company, Jefferson Hills, PA, USA) inside the PLD system at an angle of 45° normal to the target, while the RF magnetron gun was focused 45° towards the substrate using an Al_2_O_3_ target as a functionalizing/dopant material. During deposition, the fluence of the incident pulsed beam was varied from 6 to 8 J/cm^−2^ at a repetition rate of 1–10 Hz. The target-to-substrate distance was varied from 5 to 9 cm. Prior to the deposition, the Si and MgO substrates were cleaned using the standard RCA cleaning technique involving ethanol, methanol, and acetone. Initially the substrates were degassed at 750 °C to remove native oxides and other contaminants that were on the surface of the substrate. Annealing/heating of substrate is normally performed on the rear side of the sample and is monitored by a thermocouple attached to the heater set-up. The ZnO target was ablated for several minutes to remove contaminants on the surface of the target, ensuring a fresh surface prior to deposition. The RF gun was run for a few minutes to clean its target material.

The ZnO and 2D/3D ZnO heteronanostructures were either grown directly into the substrate, or grown via seed layering/templating. A seed layer in the form of a thin film affects the quality and properties of the grown nanostructure and thus improves adhesion of the grown nanostructures to the substrate. This is very essential in growing nanomaterials with applications in gas sensing. The ZnO seed layers were grown on a substrate at 300 °C for 3 min at ~10 milliTorr (~1.3 Pa) O_2_ background, while the nanostructures were grown on a substrate at a temperature between 550–650 °C and with a background pressure of O_2_, Ar, or mixed Ar + O_2_ gas. During depositions, the flow rates of THE Ar and O_2_ gases were adjusted to a total pressure from 100 milli- to 2 Torr (13 to 267 Pa) using the MKS mass flow controllers. The grown nanostructures were then cooled at a rate of 20°/min under 2 Torr (267 Pa) of Ar background.

The grown nanostructures were characterized by scanning electron microscopy (SEM) equipped with an OXFORD energy-dispersive X-ray (EDX) system (JEOL JEM-7600F, Akishima, Tokyo, Japan), operated at 30 kV; field-emission transmission electron microscopy (JEOL JEM-2100F TEM equipped with EDX, Tokyo, Japan) operated at 200 kV; X-ray diffraction (XRD, Xpert PRO PANanalytical, Almelo, The Netherlands) and X-ray photospectroscopy (XPS) (JEOL XPS-9030, Japan), performed using the Mg Kα X-ray (1253.6 eV) operated at 120 W at 0.1 eV step energy, 10 eV pass energy, and at a dwell time of 100 ms. The binding energies in XPS were corrected using the C 1s and Au 4f, while the curve fitting was performed using the SPECSURF software of the XPS-9030. Photoluminescence (PL) (SOLAR Laser Systems LQ 129, Harrietsham, UK) was used to obtain the PL spectra using the third harmonic of the pulsed Nd:YAG laser system, with an excitation wavelength of 355 nm, a pulse duration of 10 ns, and a repetition rate of 10 Hz. 

## 3. Results and Discussion

Figure 2 shows the SEM images of ZnO nanostructures grown at different PLD and RFMS parameters. All scale bars are 500 nm in length for the sake of comparison. Figure 2a shows pure ZnO nanorods, grown only by PLD at 650 °C under 500 milliTorr (67 Pa) of pure O_2_ background on a templated Si (111) substrate. The vertically grown nanorods have good aspect–height ratios. As the growth of nanostructures by PLD process is influenced by many factors, such as laser energy fluence, substrate-to-target distance, background gas and gas pressure, and substrate temperature, the crystallographic orientation of the grown film is mainly attributed to modifications in its stoichiometry and/or to the kinetic energies of the arriving ablated species [18,19]. In a normal synthesis process, the self-nucleation of the ablated species in the templated/oxidized substrate could result in the growth of randomly oriented nanostructures [20,21]. In the ZnO wurtzite structure, the three competing planes are mainly the {0001} (c-planes), {101 ®0} (m-planes), {112 ®0} (a-planes), together with the (101 ®2) directions [11]. At temperatures above 650 °C, the ablated ZnO species are said to be thermodynamically and energetically active. These qualtiies stimulate growth in the [0001] direction, or in the {0002} c-planes, being the preferential growth direction with lowest free-energy for ZnO [21]. Therefore, ZnO nanorods are typically grown at these PLD parameters.

Figure 2b shows an SEM image of the ZnO nanostructures grown by PLD with ZnO target at 650 °C, under 300 milliTorr (40 Pa) Ar + O_2_ background (190:8 sccm ratio), on Si (111) and MgO <0001> substrates. The Ar-rich environment resulted in ZnO structures with a nanoworm/nanowall-like feature. Clearly a change in the background gas at different pressure eventually changed the morphological feature of the resulting ZnO with the growth of other competing phases in the lateral direction [11]. Although other factors may have contributed also to this wall-like feature, the crystallographic orientation of the structures as scanned by XRD (shown in the succeeding section) remains that of a ZnO (0002) Wurzite structure. 

In the hybrid PLD-RFMS system, the background Ar and O_2_ gases can be simultaneously mixed and controlled by mass flow controllers. A certain gas mixture is chosen so that there is a balance in the flume shapes of generated mixed plasmas reaching the surface of the substrate. This is performed by varying the laser energy fluence of the PLD and the power of the RFMS. For the tandem system to simultaneously and effectively work, a mixture of the Ar and O_2_ gases, as well as the overall gas pressure of the chamber, must be optimized. RFMS needs an initial pressure of ~100 milliTorr (~13 Pa) for Ar gas to ignite and generate plasma, while PLD needs oxygen background to ensure ZnO film/structure with less oxygen vacancies. Figure 2c shows the ZnO nanoneedles grown by the combined PLD (ZnO target) + RFMS (Al_2_O_3_ target, 25-W power) system at 550 °C, under ~100 milliTorr (~13 Pa) Ar background on templated substrate. In this Ar-rich and O_2_-deficient environment, the ZnO <2000> is still the highly and energetically active phase and grows faster against other competing phases, which growing at slower rates. These conditions resulting in the formation of a needle-like ZnO nanostructure.

Figure 2d shows the ZnO nanostructures grown by PLD (ZnO target) + RFMS (Al_2_O_3_ target, 50 W power) on Si and MgO substrates at 650 °C under a ~150 milliTorr (~20 Pa) Ar background. When power and background Ar gas were fine and grainy, Al_2_O_3_-ZnO heteronanostructures are formed. As such, an increase in the RFMS power resulted in an increase in the number of Al_2_O_3_ species, resulting in highly energetic mixed RF and PLD plasmas that could interfere in the growth of ZnO nanorod, nanowall, and nanoneedlestructures, although other parameters, such as the shockwave generated by RFMS [16], could have prevented the growth of other structures and thus influenced the growth of fine ZnO heteronanostructures. 

The combination of PLD parameters with that of the RFMS involves a complex process, which could complicate the degree of control over the growth of the film/nanostructure. While high sputtering yield from sputtering in MS increases the collision frequency between the species, resulting therefore in a decrease in the kinetics energy of the ablated species or vice versa [16], the sizes of the plume of each respective system as well as the interface of the plume, which induce shocks to the wave formation [16], could play vital roles in the deposition of the ZnO heteronanostructure. While PLD needs a UV laser for ablation and an O_2_ background to grow ZnO nanostructures to ensure that enough O_2_ species are available for the growth of the structure, the RFMS, on the other hand, requires an inert gas, such as Argon in this case, as the sputtering gas. When the two are combined, as a result of collisions in the mixed species of PLD and RFMS plasma, extra argon ions can be created, thus affecting morphologies and deposition rates. Although the RFMS plasma can even be sustained at lower pressure of 10 milliTorr (1.3 Pa) Ar, consequently, the ZnO ions generated by PLD will be highly energetics and may even introduce re-sputtering of the grown film. On the other hand, increasing the Ar pressure to increase RFMS sputtering would, in turn, reduce the PLD plume size, thus impeding the ZnO ions from reaching the substrate. Although, the resulting sputtered species from RFMS are neutrally charged and are, therefore, unaffected by the magnetic trap in the system, combining the two systems requires the careful selection parameters to grow the Al_2_O_3_-ZnO structures, and this still remains a challenge. However, in this study, we have demonstrated that, by carefully adjusting both PLD and RFMS parameters, Al_2_O_3_ ZnO heteronanostructures can be grown.

Figure 3a shows a representative EDX spectrum of the ZnO nanostructures grown on Si (111) by PLD (target: ZnO) and RFMS (target: Al_2_O_3_) taken from TEM, showing all the expected X-ray peaks. The TEM sample was prepared using an ultrasonic bath of scratched film under ethanol conditions and likewise taking a drop into a Cu TEM grid. To determine whether the tandem system really works, i.e., whether the Al_2_O_3_ is also simultaneously deposited in the grown nanostructure, EDX analysis was performed in both SEM and TEM systems. The inset was the EDX spectrum taken from SEM, where Zn, O, and Al peaks were also clearly identified. Note that Cu and C peaks were taken from the substrate holder/stage and from the TEM grids. Since the detection limit of EDX analysis differs in the SEM from the TEM systems and depends on the elemental composition of the film/nanostructure being analyzed and the acceleration voltage of each equipment, differences in compositions are expected. The detection limit for SEM is in the range 0.1–0.5 wt%, while the TEM is in the limits of ~0.01–0.1 wt%. Therefore, SEM can be an effective technique for major and minor element analysis, while TEM can be used for trace element analysis due to its higher sensitivity. In both analyses, Al_2_O_3_ was detected in the Al_2_O_3_-ZnO structures using EDX.

Figure 3b shows the XRD patterns of ZnO nanostructures grown on Si (111) substrate with PLD + RFMS, showing the sharp peak corresponding to ZnO(002) crystallographic orientation. The inset was the XRD pattern of ZnO nanostructures grown on an MgO substrate. Other low intensity peaks could be attributed to the dominant peaks of alumina (~37.6° and ~43.7°_)_, and to the metallic Zn (39.15° and 42.2°). Interestingly, although the morphological features of the ZnO nanostructures grown at different parameters, changed quickly, the XRD spectra of all samples remarkably showed the same main prominent peaks attributed to the ZnO (002) c-axis, which is consistent with other previous results [11,12,13]. This, therefore, shows that the presence of other species did not modify/alter the texture of the ZnO growing along the 002 direction. The XRD patterns showed that the Al_2_O_3_ doping by RFMS did not structurally alter the growth direction as well, a result which is consistent with the other previously reported studies [12]. These could imply, therefore, that Al_2_O_3_ is uniformly distributed throughout the ZnO nanostructures.

Figure 4 shows the (a) Al 2p_3/2_; (b) O 1s; and (c) Zn 2p_3/2_ XPS spectra of Al_2_O_3_-ZnO nanostructures grown by PLD + RFMS. For curve fitting, all backgrounds of the spectra were determined by Shirley method and the curves were smoothed using the Savitzky–Golay method. The curve fitting is performed using the Gauss–Lorentz method within the JEOL XPS software. Figure 4a–c clearly confirm that Al, O, and Zn species were deposited in the system as verified by the previous XRD and EDX results. Although XPS scans only about 10 nm from the surface, the Al 2p_3/2_ peak is clearly located at ~72.8 eV, which, as reported by many authors, indicates the existence of the Al/AlO_x_ system [22,23,24]. A weak Al 2s peak in the XPS spectrum (figure not shown) was also noted and thus confirmed the existence of the Al 2p_3/2_ peak. The O1s spectrum in Figure 4b can be fitted with 2 peaks, which could correspond to the main oxides of ZnO and shoulder peaks attributed to the oxides of Al_2_O_3_ bonded in the ZnO, as well as to the other various carboxyl groups. The O1s peak located at 531.7 eV corresponds to oxygen in the ZnO matrix, which was reported both for the Al_2_O_3_-ZnO that was prepared by sputtering [25] and for pure ZnO [26,27]. This result therefore agrees with XRD data, making ZnO the main phase in the deposited film. The Zn 2p_3/2_ peak (Figure 4c) is located at 1023 eV, which has shifted at ~2 eV compared to metallic zinc samples. This large shift is indicative of a greater degree of oxidation for the Zn atoms in the lattice [28]. Since the XRD patterns have shown that the ZnO (002) is the main peak, while morphologies have clearly changed, the results supported the idea that Al_2_O_3_ did not doped structurally into the ZnO, as the structural feature of the ZnO remains Wurzite. Therefore, Al_2_O_3_ could have probably grown/been plated on the outer surface/layer of the ZnO, without necessarily altering the structure of ZnO. Clearly, Al_2_O_3_ is functionalized in the surface the ZnO nanostructures as verified by EDX and XPS. These results, further, demonstrated and confirmed the ability of the hybrid system of PLD and RFMS to simultaneously deposit and grow heteronanostructure ZnO materials.

Figure 5 shows the PL spectra of (a) the Al_2_O_3_-ZnO nanostructures grown by PLD + RFMS and (b) ZnO. The PL spectra of ZnO nanostructures have exhibited similar patterns (not shown) in spite of their differences in morphologies, as observed previously in the SEM scanning. The representative PL spectrum of ZnO is characterized by a sharp emission peak at around 375 nm, which is typically attributed to the band-to-band recombination, while the broad yellow-green peak, which centered at ~540 nm, is attributed to deep-level trap states, such as interstitials (Z_ni_) and vacancies (V_o_ and V_Zn_) located in the middle of the bandgap [11]. As shown in Figure 5, the undoped ZnO spectrum shows a sharper UV peak with much lower blue-green emissions, which denotes a very low concentration of defects. On the other hand, in Figure 5b the Al_2_O_3_-doped ZnO pattern shows a higher and broader peak in the green region, signifying an increase in the number concentration of defects that could be attributed to some lattice deformation brought about by the additional power from RF sputtering, resulting in more Al ions in the plasma. The presence of these could have increased the V_o_, Z_ni_, and V_Zn_. Interestingly, a weak shoulder peak at around 440 nm can be observed, which could be attributed to aluminum interstitials [29]. This results further shows the feasibility of the RFMS and PLD system as an effective tool for doping and designing ZnO nanostructures.

Further, these PL data also clarify that, although the XRD shows the same crystallographic orientation of the ZnO (0002) for all samples, the PL spectra can vary significantly. This occurs because Al doping can create additional energy levels in the bandgap which have the potential to contribute to PL emissions, as observed here in the Al_2_O_3_-doped samples. The effect of doping level, besides the different SEM features of the grown Al_2_O_3_-ZnO heteronanostructures, exhibits different PLD signature, which could affect the system’s gas sensing properties, as will be discussed below.

To explore the viability of the hybrid system, the grown Al_2_O_3_-ZnO nanostructures were deposited on Au electrodes patterned into an alumina substrate, and the resulting gas sensors were tested inside a quartz tube furnace. This measured their responses in the temperature range of 200–450 °C to CO gas, with a 50-ppm concentration in zero air. Figure 6a shows the schematic diagram of the gas sensing set-up. The gas/air mixture flow rate was fixed at 500 mL/min. The details of the measurement are reported elsewhere [30]. Figure 6b shows the CO gas response of the ZnO and Al_2_O_3_-ZnO nanostructures gas sensors as a function of operation temperature. The response, which is defined as R_air_/R_CO_ where the R_air_ and R_CO_ are the electrical resistance of the sensor in air and CO environment, respectively, increased with increasing operation temperature up to 350 °C. After this temperature, the response decreased for both sensors. This bell shape behavior could have been a result of the temperature-dependent adsorption, desorption, and gas diffusion processes involved in the sensing mechanism of metal oxide-based gas sensors [31]. However, it can be seen from the figure that the response of Al_2_O_3_-ZnO sensor was higher than that of the undoped ZnO sensor for all operating temperatures, which could be attributed to the surface morphologies, as the surface area of the doped material was higher. Further, such differences could have been attributed to the replacement of the Zn^2+^ ions by Al^3+^ ions, as this replacement can create more free electrons, leading to a more efficient conduction of the response current [32,33,34]. Consequently, the depletion region changed due to adsorption and desorption of oxygen species, and interacted with a reducing gas like CO. This played an important role in the sensing mechanism of gas sensor, as explained elsewhere [29].

The response and recovery times, defined as the required times to reach 90% of total resistance change of the sensor, were expected to decrease when the operation temperature increased. In this study, at 350 °C, both sensors showed variably fast response times (less than 8 s), whereas the recovery times were 43 s and 13 s for the undoped and Al_2_O_3_-doped sensors, respectively. It ought to be noted that the tuning of the Al concentration can give rise to different results. The gas sensing response can also be enhanced with very low concentration of Al ions, and then it decreases at higher concentrations. This latter process can affect the formation of the depletion layer [33]. The doped sample showed higher responses at higher gas concentrations, which probably occurred due to the creation of additional adsorption sites in the Al-doped sensor; however, further study is still needed to elucidate these phenomena. 

The mechanism for the growth of heteronanostructures, which involves combining plumes of PLD and RFMS resulting into mixed-species plasma, is quite complex, and until now has been unclear and even conflicting. Explanations could only be attempted based on the individual character/nature of the deposition techniques. Pluming from PLD results in shockwave formation, thereby increasing pressure and thus affecting the sputtering in RFMS. While PLD is a fast deposition process, RFMS, on the other hand, is a slow deposition technique. Species from PLD have high velocities that can tend to dislodge species in the plume from the RFMS, preventing them from reaching the substrate surface. For optimization, then, the parameters of PLD and RFMS have to be carefully chosen to enable the growth of nanostructures. In order for the heteronanostructures to grow, distances between targets from PLD and RFMS (if oriented 45° towards the substrate) should be appropriately considered so that the velocities of the mixed species can be identical to allow the nanostructures to grow [14]. In the tandem systems, one can, therefore, adjust the following parameter: (a) the frequency of pulses in PLD, in order to optimize the deposition of RFMS materials, while at the same time growing the ZnO structures; (b) and the temperature of the substrate to provide the incoming species with more mobility, simultaneously reducing the formation of droplets from the combined plumes; (c) as well as background pressure, which will increase the probability of creating ramified structures; and (d) energy fluence, as high fluence tends to vaporize the material at the substrate.

In spite of these challenges, we have demonstrated the viability of using the PLD-RFMS set-up to design and grow ZnO and Al_2_O_3_-ZnO heteronanostructures as base materials for gas sensing applications.

## 4. Conclusions

This study demonstrates the capability of a tandem system of PLD and RFMS system to design and grow Al_2_O_3_-ZnO heteronanostructures with different morphological features. This hybrid approach is used to synthesize 2D/3D ZnO and Al_2_O_3_-doped ZnO nanostructures for possible gas sensing applications. The ablation technique in PLD is combined with the sputtering technique of RFMS to create mixed-species plasma for use in the growth of nanostructures. By carefully adjusting respective parameters of the two systems, such as laser energy fluence, substrate temperature, and the pressure of background gases, a series of nanoneedles/nanospikes, nanowalls, nanorods, and grainy Al_2_O_3_ ZnO structures were grown. The characterizations of the nanostructures reveal that, although the morphological features of the ZnO nanostructures change considerably, the crystalline feature of ZnO remains Wurzite. Further, when the RF sputtering is introduced to the system, the resulting films are coarser/grainy in nature compared to the needle-like feature of the ZnO obtained with a low RFMS power. The growth of structures in the combined system could only be attempted based on the individual character/nature of the deposition techniques. As RF sputtering interferes in the growth kinetics of PLD by slowing down the ZnO ions that are ablated by PLD, the ions coming to the surface of the substrates may have lower energies, thus preventing needle-like structures from growing as occurs in this case. Therefore, designing and growing ZnO heteronanostructures revolves around the interplay of frequency of pulses in PLD, temperature of the substrate, background pressure, and laser energy fluence. 

Although the exact mechanisms involved remain under debate, the tandem of PLD and RFMS has demonstrated the capacity to grow ZnO and Al_2_O_3_-functionalized ZnO nanostructures that are exceptional with potential applications in optoelectronics, such as in bio/gas sensors.

## Figures and Tables

**Figure 1 nanomaterials-13-01345-f001:**
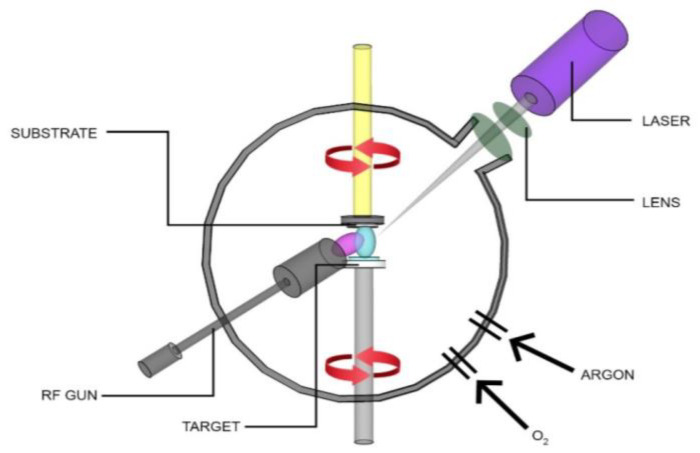
Schematic diagram of the PLD and RFMS hybrid System.

**Figure 2 nanomaterials-13-01345-f002:**
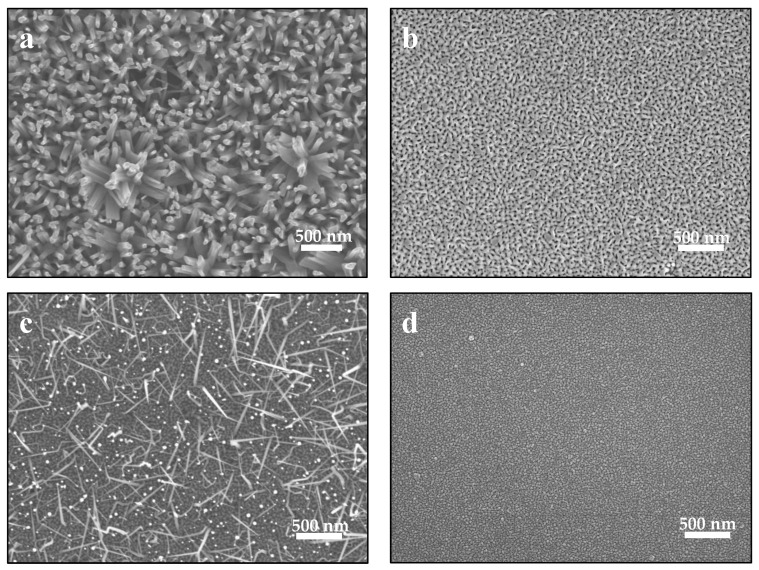
ZnO nanostructures grown by (**a**) PLD at 650 °C under 500 milliTorr (67 Pa) pure O_2_ background (nanorods); (**b**) PLD (ZnO target) at 650 °C, under 300 milliTorr (40 Pa) combined Ar + O_2_ background (190:8 sccm ratio) (worm-like); (**c**) by PLD (ZnO target) + RFMS (Al_2_O_3_ target, 25 W power) at 550 °C, under ~100 milliTorr (~13 Pa) Ar background by template method (nanoneedles); and (**d**) PLD (ZnO target) + RFMS (Al_2_O_3_ target, 50 W power) at 550 °C, under 150 milliTorr (~20 Pa) Ar background (heteronanostructures). All scale bars are of 500 nm in length for comparison.

**Figure 3 nanomaterials-13-01345-f003:**
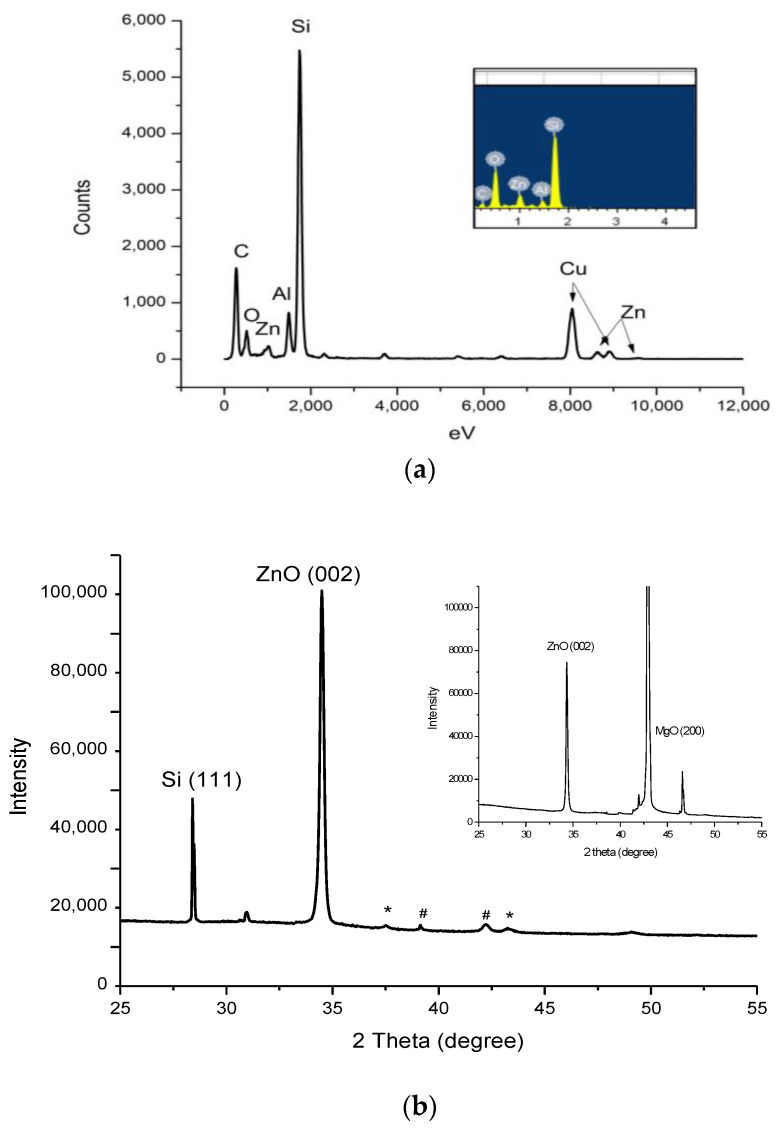
(**a**) EDX pattern of ZnO grown on Si (111) by PLD (ZnO) + RFMS (Al_2_O_3_) taken by TEM. As confirmed by EDX spectrum taken from SEM, the Zn, O, and Al peaks were clearly identified (see Inset); (**b**) XRD pattern of ZnO nanostructures grown on Si (111) substrate by PLD + RFMS, showing the sharp peak corresponding to ZnO (002). Other low intensity peaks could be attributed to alumina, denoted by (*) and to metallic Zinc, denoted by (#). Inset is the XRD pattern of the ZnO nanostructures grown on MgO substrate.

**Figure 4 nanomaterials-13-01345-f004:**
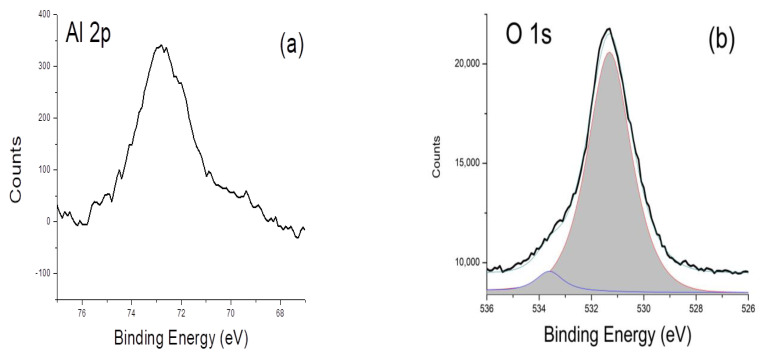
XPS spectra of Al_2_O_3_-ZnO nanostructures grown by PLD + RFMS. (**a**) Al 2p_3/2_; (**b**) O1s: (**c**) Zn 2p_3/2_ lines.

**Figure 5 nanomaterials-13-01345-f005:**
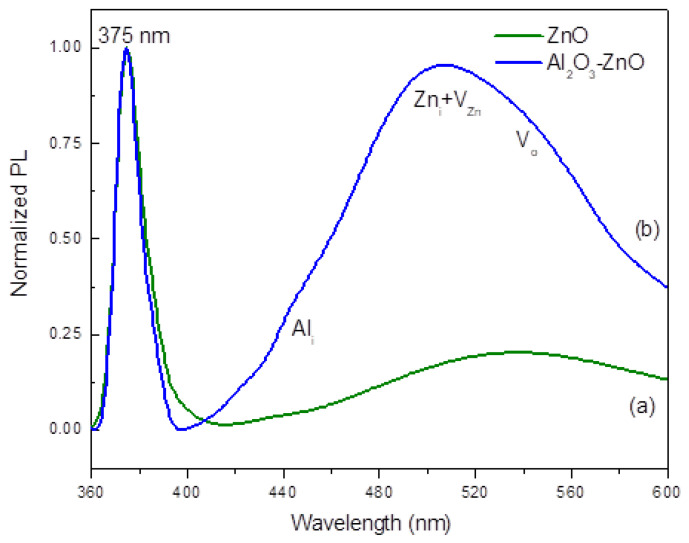
PL spectra of the (**a**) undoped ZnO (green line) and (**b**) Al_2_O_3_-ZnO nanostructures grown by PLD + RFMS (blue line).

**Figure 6 nanomaterials-13-01345-f006:**
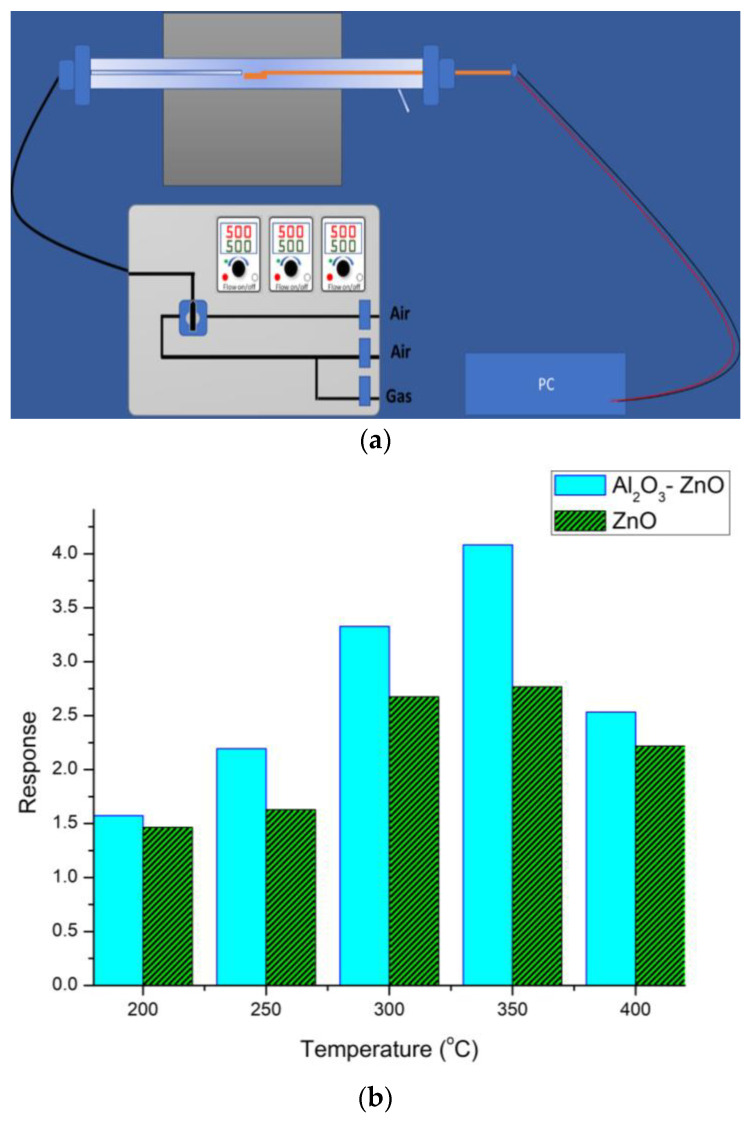
(**a**) Schematic diagram of the gas sensing measurement set-up. (**b**). Responses of Al_2_O_3_-ZnO nanostructures grown on Au-electrode Al_2_O_3_ sensors under CO gas condition at 50 ppm.

## Data Availability

Not applicable.

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
