# Peer review of "Optimization of Pulsed Laser Ablation and Radio-Frequency Sputtering Tandem System for Synthesis of 2D/3D Al2O3-ZnO Nanostructures: A Hybrid Approach to Synthesis of Nanostructures for Gas Sensing Applications"

_nanomaterials, 2023, doi:10.3390/nano13081345_

Round 1

Reviewer 1 Report

In the submitted manuscript the authors have studied an innovative approach to synthesize Al2O3-ZnO nanostructures by simultaneous deposition by pulsed laser deposition and radio frequency magnetron sputtering. the magnetic and transport properties of YMnO3 thin films (doped with Ca or Sr) grown by pulsed laser deposition. The work is well suited for the journal and interesting enough for its publication but there are several issues to be addressed before I recommend its publication. In particular, my commentaries are the following ones:

a)      Grammar mistakes: There are several typos, grammar and/or style errors. Please, check the grammar carefully. I put here several examples:

i.                    line 30: I think that “such in perovskite…” should be omitted.

ii.                  Line 60: I think that it should be “others” instead of “other”

iii.                Line 77:  I would suggest “…RFMS is a very…” instead of the current text.

iv.                Line 268: I would put “has resulted in an increase of the number…” instead of the current text.

v.                  Line 426: “interval between pulses”

vi.                Line 428: “the temperature of the substrate was/is increased…”

vii.              Line 484-486. I think that there is something missing in that phrase. I do not understand it very well.

b) Introduction: The introduction is quite long, which is not a bad thing. However, I think that more references should be added, as there are several paragraphs without any reference, which is quite odd, given the fact that the authors are commenting the previous results found in the literature. In some cases, the references have been provided but also be indicated in several parts of the introduction, not only once.

c) Results and discussion, Figure 3 and related text: No values are provided about the EDX results. It could be interesting to give some normalized values of the content of the different elements in the samples, ignoring the grid and substrate.

Additionally, there are several small peaks in the DRX pattern, which are not mentioned in the text. It is true that the ZnO (002) peak is huge compared with them, but, nevertheless, there is something else. Do some of those peaks correspond to the alumina phase?

d) Figure 5, caption: In the caption, it should be indicated that (a) corresponds to the ZnO alone and (b) to the mixture Al2O3-ZnO.

e) Conclusions: the first reference to the existence of nanospheres/nanodisks is here. If there are some of these elements, which is not so clear for me, it should clearly be indicated in the results and discussion (as it was the case for other nanostructures).

Reviewer 2 Report

There are some shortcomings which must be removed prior to publication

Introduction is pretty lengthy and should be shortened

Line 191: 10-6 Pa

Line 199: degassed “at”

Lines 201/202: What is “normally” doing here?

Line 208: what kind of seed layer?

Figure 2 and related text: I suggest to add a table with deposition techniques/parameters for each of the 4 nanostructures.  

Line 271: shockwave generated by RFMS: which kind of shockwave and where?

Paragraph starting line 289: what does it tell us? I suggest to remove this part.

Figure 3: Sure that it is Si(100) and not Si(111) reflection? Caption: ZnO(2000) ? What is the crystalline phase? 

Lines 340/341: background subtraction. In my opinion, no background is subtracted from the signal in Figures 4(b) and (c). Vertical axis of Figure 4(a) should start close to “0”.

Line 355: how was the energy axis calibrated?

Figure 5: (a) and (b) are not mentioned in the caption. Colour: blue and green are difficult to distinguish, use red for green or blue instead.

Line 395: why is it due to the “additional power” rather than to the defects due to Al atoms?   

Round 2

Reviewer 2 Report

The paper is improved and now ready for publication.